# Thai *Cannabis sativa* Leaves as a Functional Ingredient for Quality Improvement and Lactic Acid Bacterial Growth Enhancement in Kombucha

**DOI:** 10.3390/foods14060942

**Published:** 2025-03-10

**Authors:** Qurrata A’yuni, Kevin Mok, Massalin Nakphaichit, Kamolwan Jangchud, Tantawan Pirak

**Affiliations:** 1Department of Product Development, Faculty of Agro-Industry, Kasetsart University, 50 Ngamwomgwan, Lat Yao, Chatuchak, Bangkok 10900, Thailand; qurrata.a@ku.th (Q.A.); fagikwj@ku.ac.th (K.J.); 2Department of Biotechnology, Faculty of Agro-Industry, Kasetsart University, 50 Ngamwomgwan, Lat Yao, Chatuchak, Bangkok 10900, Thailand; kevin.m@ku.th (K.M.); fagimln@ku.ac.th (M.N.); 3Specialized Research Unit: Prebiotics and Probiotics for Health, Department of Biotechnology, Faculty of Agro-Industry, Kasetsart University, 50 Ngamwomgwan, Lat Yao, Chatuchak, Bangkok 10900, Thailand

**Keywords:** fermentation, kombucha, metagenome, cannabis leaves, lactic acid bacteria

## Abstract

Kombucha is a well-known fermented drink that gained interest due to its gut health benefits. However, it has a harsh taste of acetic acid and is hard to consume. Thai Cannabis leaves (*Cannabis sativa* sp. Hang Kra Rog Phu phan ST1) contain high protein and phytochemicals which can improve the growth of lactic acid bacteria (LAB) and enhance the organoleptic quality of the Kombucha. This study revealed the effect of infusing assam green tea leaves with cannabis leaves on the fermentation rate, microbial communities, volatile compounds, and overall quality and taste of the kombucha. The high protein content (23.10%) of Cannabis leaves was found. Phytonutrients and phytochemicals found in the leaves promotes LAB growth, which resulted in the higher number of LAB in the treatment with cannabis leaves. At the end of fermentation (day 7), the highest LAB count (5.53 log CFU mL^−1^) was presented in kombucha infused with 30% cannabis leaves. Kombucha with better quality, higher pH, and less acidity was obtained in a dose manner. The change in microbial communities was detected using metagenomic analysis. The prominence of *Dekkera* and *Komagataeibacter*, with low abundance of *Zygosaccharomyces* and *Weissella* were identified. These microorganisms improved flavor by lessening strong fermented odor and harsh acidic taste. From volatile compounds, HS-SPME-GCMS revealed that kombucha infused with 30% cannabis leaves possessed less acetic acid, ethanol, and carbon dioxide and gave a better odor and taste. Hence, cannabis leaves was the novel substrate for kombucha fermentation by enhancing LAB growth and improving the overall qualities.

## 1. Introduction

Kombucha is a functional fermented drink recognized for its acetic taste, natural fizziness, and the formation of a cellulose biofilm through acetic acid fermentation. It has gained popularity due to its perceived health benefits, which include antioxidants, antimicrobial, antidiabetic, and digestive health properties [1,2,3,4]. Traditional kombucha is typically fermented with tea-based substrates (e.g., black, fu-brick, pu-erh, green, and rooibos) using symbiotic bacteria and yeast cultures (SCOBY) for 7–14 days. The use of various tea substrates, starter cultures, and fermentation conditions (e.g., time, temperature, and incubation methods) can change kombucha’s overall qualities [5,6]. In response to this, traditional kombucha often leads to inconsistent product quality, such as high acidity levels, strong alcohol content, and harsh acetic acid odors [7]. It also often has a low abundance or absence of viable lactic acid bacteria (LAB), which are crucial for enhancing the flavor appeal and health benefits [8,9]. These hurdles highlight the need for further studies to improve kombucha’s quality and functionality.

Recent studies have shown that fermenting kombucha with herbs and leaves improved its quality and functional properties [10,11,12]. During fermentation, polyphenols in plants are used as substrates and hydrolysis by polyphenol-associated enzymes occurs, which resulted in the presence of small compounds of polyphenols such as gallic acid, quercetin, and kaempferol. These compounds can promote the growth of beneficial LAB while inhibiting the growth of pathogenic bacteria during the fermentation process [13]. In addition, lime leaves enhanced the signature flavor of kombucha [14], while the mint leaves escalated the desired ester compounds and achieved the highest overall sensorial acceptance [12]. In terms of bioactivities, kombucha fermented with wax mallow and African mustard leaves increased polyphenol, antioxidant, and antimicrobial properties [10,15]. In this regard, fermenting kombucha with alternative substrates provides nutritional and functional ingredients, thus influencing the biochemical pathway, fermentation conditions, and the growth of microbial communities [16]. Xu et al. [17] highlighted that fermenting kombucha with alternative substrates generated different microbial communities and diversity greater than that of tea substrates. This can vary the chemical composition generated during fermentation, including acetic acid, ethanol, valeric acid, propionic acid, lactic acid, linalool, and 2-methyl propanoic acid [16,17]. These quality changes are driven by the dominant yeast and bacteria present, indicating that using an alternative substrate affects the dynamic microbiota growth [18]. In particular, the effect of the different substrates has evolved the presence of certain strains, such as LAB (e.g., *Oenococcus oeni*, *Lactobacillus nagelii*) during kombucha fermentation [18]. These strains have been crucial in modulating microbial communities. As a result, kombucha’s qualities were characterized by softening vinegar acidity, enhancing fruity flavor, managing the sugar content, and improving antioxidant activity [17,18,19,20]. Among the various alternative substrates used for kombucha fermentation, none have been reported to promote LAB growth naturally. Cannabis (*Cannabis sativa* L.) is one of the medicinal plants providing an abundance of bioactive compounds in leaves. The heritage phytochemical compounds from cannabis leaves are cannabinoids, specifically cannabidiol (CBD) and delta-9-tetrahydrocannabinol (THC). It also contains unique polyphenol compounds (cannflavin and cannabisin) and is rich in terpene compounds. These compounds provide functional properties such as flavoring substances, nutrient sources, and various biological properties [21,22,23]. Among the various species, *Cannabis sativa* sp. Hang Kra Rog Phu Phan ST1 is cultivated in Thailand, and it contains an abundance of phytochemicals with high CBD and low THC compared to other species [24,25]. According to Jangchud et al. [26], Thai cannabis leaves contain high protein (20.86% db), high in crude fiber (12.40% db), and fat (3.57% db), offering phytonutrients as the growth promoter for microbial fermentation. Several studies reported the merit of cannabis leaves for their functionality since their legal status has evolved. Moreover, cannabis leaves have been used as dietary supplements for the prevention of chronic degenerative diseases [23]. Furthermore, pure extracts of cannabis leaves have been shown to enhance the growth of *Bifidobacterium* and *Lactobacillus*, which could be due to cannabinoid compounds acting as antimicrobial activity to inhibit pathogenic microbes [22,27]. Cannabinoids such as CBD and THC have been reported to induce the growth of beneficial bacteria like *Bifidobacterium* and *Lactobacillus* [22]. One proposed mechanism is that cannabinoids exhibit antimicrobial properties, selectively inhibiting pathogenic microbes and might create a more favorable environment for LAB growth. This aligns with Blaskovich et al. [28], who demonstrated that CBD exhibits potent antibacterial activity against biofilms and pathogenic bacteria.

Recently, there has been growing interest in formulating cannabis leaf-infused beverages to enhance their quality and functional benefits [23,29]. The Global MINTEL database has identified 92 cannabis drinks launched globally between 2019 and 2024 [29]. The cannabis products expanded to various categories of cannabis drinks, including energy drinks, carbonated soft drinks, juice drinks, nutritional drinks, hot beverages, alcoholic beverages, and water drinks [29]. When used for product applications, cannabis leaves were infused through direct addition, steeping in hot water, or adding in the form of extract, powder, capsules, and emulsion [23]. In Thailand, cannabis leaves have emerged as a novel functional ingredient, with legal requirements stipulating that the THC and CBD content in the final product must not exceed 1.60 mg and 1.41 mg per serving, respectively [30]. Thai cannabis leaves have been applied as bioactive ingredients formulated in functional drinks to enhance their functionality and bioactivity [31]. Jangchud et al. [26] emphasized that incorporating the powdered cannabis leaves into the product increased total polyphenol content and antioxidant activity. Other studies have pointed out the benefits of cannabis application, as follows: increasing protein and essential micronutrient contents, enhancing the bioactivity and therapeutic effects, improving gut health, and leveling flavor satisfaction [21,23,29]. Thus, cannabis leaf is proposed as a novel substrate to be infused during kombucha fermentation. We also monitored the effects of cannabis leaf concentration and fermentation time on lactic acid bacteria growth and the changes in quality characteristics. Additionally, the effect of cannabis leaf infusion on the changes in volatile compounds and microbial communities using metagenomics analysis in the final fermented kombucha was also analyzed to reveal a better understanding of phytochemical’s impact on kombucha quality.

## 2. Materials and Methods

### 2.1. Plant Materials and Chemical Reagents

Dried assam green tea leaves (*Camellia sinensis* var. *assamica*) were purchased from a tea company located in Chiang Rai, Thailand. Dried cannabis leaves (*Cannabis sativa* sp. Hang Kra Rog Phu phan ST1) were obtained from Chalermphrakiat Sakon Nakhon, Kasetsart University, Bangkok, Thailand. The symbiotic culture of bacteria and yeast (SCOBY) was sourced from Swan Ketogenic Co., Ltd., Bangkok, Thailand, and was used for kombucha fermentation. The chemical reagents used in this study included gallic acid, Folin–Ciocalteu reagent, 2,2-diphenyl-1-picrylhydrazyl (DPPH), 6-hydroxy-2,5,7,8-tetramethylchroman-2-carboxylic acid (Trolox), and bromocresol purple. All the chemicals used in this experiment were purchased from Sigma-Aldrich (St. Louis, MI, USA). Furthermore, ethanol (95%, analytical grade) was procured from Qrec (Auckland, New Zealand).

### 2.2. Analysis of Antioxidant Properties and Proximate Composition of Plant Materials

#### 2.2.1. Antioxidant Properties of Plant Materials

The antioxidant properties of assam green tea and cannabis leaves were investigated. Each leaf was infused following the method described by Tongsai et al. [32] to obtain the infused tea for further analysis. The total phenolic content (TPC) was analyzed using the Folin–Ciocalteu method, while antioxidant capacity was determined using the DPPH radical assay. These assays were performed in triplicate for each infused tea, following the method of Rahmani et al. [10]. The results for TPC and antioxidant capacity were expressed as milligrams of gallic acid equivalents per milliliter (mg GAE mL^−1^) and milligrams of Trolox equivalents per milliliter (mg TE mL^−1^), respectively.

#### 2.2.2. Proximate Analysis of Plant Materials

The proximate composition of assam green tea and cannabis leaves was determined using the method of the Association of Analytical Chemists (AOAC, 2023) [33], involving moisture content, ash, fat, protein, fiber, and carbohydrate (by difference), and the results were presented on a g/100 g wet weight basis.

### 2.3. Preparation of Cannabis Leaf-Infused Kombucha

In this experiment, two factors were studied: (i) the fermentation time (0, 3, 5, and 7 days) and (ii) the concentration of cannabis leaves added to formulated kombucha tea (0%, 15%, 30%, and 50%). Briefly, the formulations were prepared with varying ratios of dried assam green tea to cannabis leaves (%): 100:0 (KACL0), 85:15 (KACL15), 70:30 (KACL30), and 50:50 (KACL50). The kombucha treatments were investigated using a 4 × 4 factorial randomized design to obtain 16 treatments (Figure 1). The abbreviations for each treatment are presented in Figure 1.

The infused tea was prepared following the formulation shown in Figure 1 by weighing dried assam green tea and cannabis leaves to achieve a final concentration of 10 g L^−1^ (*w*/*v*). A sucrose solution (120 g L^−1^, *w*/*v*) was prepared by dissolving sucrose in 1 L of hot water in a sterilized glass vessel. Each concentration of formulated cannabis leaves was infused with tea using a magnetic hot plate (IKAC-MAG HS 7, IKA Works Guangzhou, China) at 95 ± 2 °C for 15 min, according to a previous method [32]. After infusion, the resulting tea was filtered through cheesecloth to remove the leaves, then transferred to glass bottles, and sterilized by autoclaving at 121 °C for 15 min. The sterilized cannabis-infused tea was allowed to cool to room temperature. Subsequently, the SCOBY culture (64 mL L^−1^, *v*/*v*) was inoculated into the sterilized cannabis-infused tea (1 L) in glass vessels prepared according to the formulations described in Figure 1. The kombucha fermentation was conducted at 25 ± 1 °C in a standing incubator (Sanyo Incubator MIR 523; Thanes Development Co., Ltd., Bangkok, Thailand) for seven days. The samples were collected on days 0, 3, 5, and 7 of fermentation to analyze changes in the overall quality and microbial characteristics.

### 2.4. Effect of Cannabis Leaves Concentration and Fermentation Time on the Changes in Microbial, Physicochemical, and Antioxidant Properties of the Fermented Kombucha

In this experiment, the samples obtained from Section 2.3 were subject to the analysis of various properties, as shown below.

#### 2.4.1. Microbiological Analysis

The microbial enumeration was performed to determine the total yeast and total LAB counts in 16 kombucha samples, following a previously described method with slight modifications [2]. For yeast analysis, Potato Dextrose Agar (PDA) (Himedia, Thane, India) was prepared with distilled water and 1% tartaric acid. For LAB analysis, De Man, Rogosa, and Sharpe agar (MRS) (Difco, Detroit, MI, USA) was prepared with distilled water and supplemented with 0.005% bromocresol purple. Samples were subjected to serial dilutions (1:9, *v*/*v*) in peptone water, ranging from 10^−1^ to 10^−7^. The media and peptone water were sterilized by autoclaving at 121 °C for 15 min. Each diluted kombucha sample (1 mL) was plated onto Petri dishes containing PDA (15 mL) for yeast enumeration and MRS agar for LAB analysis using the pour plate method. After media solidification, PDA plates were incubated at 25 °C for four days, and MRS plates were incubated at 35 °C for three days. Microbial counts were reported as log colony-forming units per milliliter (log CFU mL^−1^).

##### 2.4.2. pH Value and Total Acidity

The pH and acidity of the 16 kombucha samples were analyzed in triplicate by following the method proposed by Wang et al. [34]. The pH was measured using a digital pH meter (MODEL 9625, HORIBA, Kyoto, Japan). Total acidity was determined using a titrimetric method and expressed as g L^−1^ of acetic acid.

#### 2.4.3. Total Soluble Solids and Turbidity

The total soluble solids of the 16 kombucha samples were measured in triplicate using a hand refractometer (MASTER-53M; Atago, Tokyo, Japan) at room temperature, following the method described by Wang et al. [34]. The data were expressed as the soluble solids content (°Brix). For turbidity analysis, 1 mL of each sample was centrifuged at (4200× *g* for 10 min at room temperature) following the method revealed by Wang et al. [35]. Turbidity values were measured in triplicate at 660 nm using a BIOMATE 3S UV-Visible Spectrophotometer (Thermo Fisher Scientific, Waltham, MA, USA).

#### 2.4.4. Total Phenolic Content and Antioxidant Capacity

The polyphenol content of the 16 kombucha samples was determined in triplicate using the Folin–Ciocalteu method with a modification [10]. The sample (100 μL) was mixed with Folin–Ciocalteu reagent 0.2 N (500 μL) and sodium carbonate solution 7.5% (400 μL) was then added to a reaction mixture in a 2 mL test tube. The mixture solution was incubated for 25 min at room temperature. The absorbance of the mixture solution was measured at 765 nm using a UV-spectrophotometer (BIOMATE 3S UV-Vis; Thermo Fisher Scientific Co., Ltd.; USA). A standard calibration curve was generated using gallic acid (5 μM to 50 μM; R^2^ = 0.9987). The results were reported in milligrams of gallic acid equivalents per milliliter (mg GAE/mL).

The DPPH antioxidant capacity of the 16 kombucha samples were measured in triplicate according to the method proposed by Rahmani et al. [10] with a modification. The kombucha sample (200 μL) was mixed with DPPH ethanolic solution 0.1 mM (800 μL) in a 2 mL test tube. After thorough vortexing for homogeneity, the mixture solution was incubated for 25 min at room temperature in a dark room. The absorbance of the samples was then measured at 517 nm using a UV-spectrophotometer (BIOMATE 3S UV-Vis; Thermo Fisher Scientific Co., Ltd.; USA). A standard Trolox curve was determined with concentrations ranging from 10 μM to 140 μM (R^2^ = 0.9987). The results were expressed in milligrams of Trolox per milliliter (mg TE/mL).

### 2.5. Microbiota Diversity Using Metagenomic Analysis

The sample selected for further analysis in this experiment was from the sample analyzed previously (in Section 2.4), regarding the prominent properties and the large number of lactic acid bacterial count. Metagenomic analysis was performed on cannabis-infused kombucha (KACL30-D7) and kombucha tea (KACL0-D7) following previously described protocols [36,37]. The microbial DNA was isolated using the QIAamp Fast DNA Stool Mini kit (Qiagen, Hilden, Germany), according to the manufacturer’s protocol. The quantity and quality extracted DNA were evaluated using a NanoDrop 2000c spectrophotometer (Thermo Scientific, USA) and visualized using 1.2% (*w*/*v*) agarose gel electrophoresis and SYBR Green dye. A total of 16s rRNA and ITS2 amplicon sequencing were performed on the Illumina Novaseq 6000 (San Diego, CA, USA). The bioinformatics analysis was conducted to identify the microbial community with taxonomic profiles. The USEARCH software (version 11.0.667) was used to merge the sequencing reads. The primer sequences were trimmed using the search_pcr2 algorithm. Subsequently, the trimmed sequences were subjected to quality filtering, whereby sequences shorter than 350 bp for 16S rRNA sequences and 250 bp for ITS2 amplicons were removed. Additionally, sequences with an expected error ≥0.05 were excluded. The operational taxonomic units (OTUs) were clustered within a 97% similarity threshold, and the sintax algorithm was used to determine microbial taxonomies. The database for fungi identification was derived from UNITE  +  INSD dataset for fungi version 8.3 10.05.2021 [38], while for bacteria, Ribosomal Database Project (RDP) training set v18 database was used [39]. Alpha diversity was assessed using the Shannon, Evenness, and Chao1 Index.

### 2.6. Volatile Compounds Using HS-SPME-GC-MS Method

Volatile compound analysis was conducted using the method of Wang et al. [40] on the selected sample from the previous experiment. The volatile compounds were identified using headspace solid-phase microextraction gas chromatography–mass spectrometry (HS-SPME-GC-MS) with a TriPlus RSH autosampler. Samples (5 mL) were placed in a 20 mL screw-capped headspace vial, and a Solid Phase Micro-Extraction (SPME) fiber was introduced to capture the gas vapor within the vessel. The samples were agitated at 500 rpm for 10 min, followed by continuous stirring at 1000 rpm and heating at 70 °C for 30 min to entrap the volatile compounds. The absorbed volatile compounds of both samples were introduced from the SPME fiber into the GC–MS-TRACE1300 system (Thermo Scientific, USA) at the injection port, set at 250 °C for 4 min. Helium gas (99.9% purity) served as the carrier gas, facilitating the transport of volatile compounds to a TG-5MS column (30 m × 0.25 mm × 0.25 μm) at a flow rate of 1 mL min^−1^. The column temperature was initially set at 50 °C (held for 4 min), then increased to 290 °C (held for 5 min) at a flow rate of 6 °C min-1, and further increased to a final temperature of 300 °C at a flow rate of 10 °C min^−1^ (held for 5 min). The volatile compounds were identified by analyzing the mass spectra of the analytes and comparing the obtained data with the National Institute of Standards and Technology (NIST). The results are expressed as a percentage of the peak abundance.

### 2.7. Statistical Analysis

Two-way analysis of variance (ANOVA) was used to evaluate the significant differences in each sample that were affected by the concentration of cannabis leaves and fermentation times. Duncan’s multiple range test (DMRT) was applied when a significant effect was detected (*p* < 0.05). Pearson correlation coefficients (r) were calculated to determine the correlations among the variables during kombucha fermentation. Statistical analysis was performed using ANOVA, DMRT, and Pearson correlation using SPSS statistics software version 22.0 for Windows (Thaisoftup Co., Ltd., Bangkok, Thailand). Principal component analysis (PCA) and hierarchical cluster analysis (HCA) were performed using XLSTAT version 19.6 software (Addinsoft, New York, NY, USA). Venn diagrams were generated using the jVenn online tools.

## 3. Results

### 3.1. Antioxidant Properties and Proximate Compositions of Plant Materials

It was found that assam green tea exhibited a TPC of 10.8 times higher and DPPH antioxidant capacity of 33.78 times higher when compared to those of cannabis leaves (Table 1). This disparity can be attributed to the high catechin content typically found in tea leaves, particularly epigallocatechin gallate (EGCG). This compound has strong antioxidant properties owing to its hydroxyl and galloyl groups, making EGCG more effective against the reactive oxygen species present in tea leaves [32,41].

When considering the proximate compositions, assam green tea leaves exhibited higher amounts of carbohydrates and crude fiber, whereas cannabis leaves had higher levels of crude fat, ash, and protein (Table 1). These results corresponded with the study by Jangchud et al. [26], highlighting that Thai cannabis leaves contain high protein and fiber content, and revealed that phytonutrients in these two plants were able to be a substrate for microbes during kombucha fermentation [9,18].

### 3.2. Effect of the Concentration of Cannabis Leaves and Fermentation Time on the Growth of Lactic Acid Bacteria (LAB) and Total Yeast and Mold (TYM) of the Fermented Kombucha

The effect of the concentration of cannabis leaves infused into kombucha significantly increased the number of LAB and TYM as the fermentation progressed (Figure 2). These results were consistent with a study by Xu et al. [17]. When considering the impact of fermentation time, the number of LAB and TYM across the kombucha samples increased significantly by 4 and 2 log cycles, respectively, by day 3 of fermentation. The fermenting microbiota in SCOBY cultures, particularly yeast (e.g., *Saccharomyces*, *Dekkera*, *Zygosaccharomyces*), utilize carbon sources as nutrients to induce their growth [42].

As the fermentation continued, the highest number of TYM (7.04 log CFU mL^−1^) and LAB (5.53 log CFU mL^−1^) was revealed in KACL50-D7 and KACL30-D7, respectively. On day 7 of the fermentation, a decrease in LAB number was observed in all samples except for KACL30-D7, while a decrease in TYM counts was elucidated in KACL0-D7 (control). These findings suggest a synergistic interaction between cannabis leaf and microbial growth, primarily by maintaining a suitable pH for sustained microbial activity and consisting of the soluble phytonutrients such as polyphenols as a beneficial substrate. The growth of *Lactobacillus* strains was enhanced in the dose manner effect (Figure 2b). As a result, the increased production of lactic acid led to a decrease in pH and bacteriocins might also be produced, which may have inhibited pathogenic microorganisms [43]. Moreover, the phytonutrients obtained from plant materials may accelerate fermentation rates to support microbial growth [44].

Moreover, a positive correlation between the number of yeast and LAB (r = 0.971; *p* < 0.01) was observed, indicating microbial cooperative interactions during kombucha fermentation. This phenomenon occurs when yeast provides the monosaccharides and ethanol to induce the growth of LAB and acetic acid bacteria (AAB), respectively, in order to produce organic acids [45]. This manufactured acidic condition was able to inhibit the growth of microbial spoilage and pathogens. On the other hand, the viability of the non-acid-tolerant yeasts decreased (Figure 2a). In other words, the number of LAB decreased on day 7 due to the resulting higher ethanol concentrations from yeast metabolism (Figure 2b), because the toxic environment was created [7,45]. In addition, the number of TYM obtained in cannabis leaf–kombucha samples consistently increased until the final fermentation. This could be due to the remaining nutrients that were able to promote the growth of certain yeasts, specifically *Brettanomyces* or *Dekkera*, which are recognized as acid-tolerant and able to utilize residual sugars such as glucose and fructose [46].

Furthermore, the obtained number of TYM and LAB in the current study agreed with previous findings, reaching 6–7 log cycles [11,47], and exceeding 3 log cycles, respectively [11]. Typically, *Lactobacilli* are present in either low abundance or absence in fermented kombucha described in previous studies [8,11,47]. Hence, this study indicated that LAB viability can be naturally enhanced in kombucha without the addition of probiotic cultures, as shown in previous studies [48,49]. Nissen et al. [50] emphasized that plant-based hemp drinks provide an effective medium for fermentation, along with a strong prebiotic activity to promote probiotic growth. Thus, cannabis leaves are suggested as a novel functional ingredient for promoting microbial growth during fermentation.

### 3.3. Effect of the Concentration of Cannabis Leaves and Fermentation Time on Quality Characteristics Changes in the Fermented Kombucha

The pH of kombucha was increased by the addition of cannabis leaves. However, it gradually declined over extended fermentation times, despite the effect of cannabis concentration (Figure 3a). The pH levels across cannabis-infused kombucha were 3.27–3.74, whereas the pH in kombucha tea was 3.10–3.55 throughout fermentation. Among the various samples, the highest pH was achieved by KACL50-D0, which could be affected by the higher concentration of cannabis leaves, demonstrating the fermentation rate might occur more slowly. In the current study, the obtained pH value ranges were suitable to promote the growth rate of fermenting microbiota, including yeast, AAB, and LAB during kombucha fermentation. This corresponded with recent studies [18,47], which revealed the suggesting pH ranges were in the range of 2.5–4.2. These pH ranges also ensured the kombucha remained safe for consumption and prevented the growth of enteropathogenic microorganisms [46].

Furthermore, the strongest negative correlation (r = −0.847; *p* < 0.01) between pH and acidity was found, which showed that a decrease in pH aligned with an increase in total acidity. These significant changes in pH and acidity were observed in the kombucha samples from days 5 to 7 of fermentation (Figure 3b). This could be attributed to the metabolism of AAB strains (e.g., *Acetobacter*, *Gluconobacter*, *Komagataeibacter*) during acetic acid production [7]. Hence, the highest acidity was observed in KACL0-D7 (4.75 g L^−1^). The cannabis leaf-infused samples obtained lower acidity levels, specifically in KACL30-D7 and KACL50-D7 (4.31–4.42 g L^−1^). These findings highlighted that cannabis-infusion could slow down the rate of ethanol oxidation due to the lower carbohydrate content. This is in agreement with a previous study which reported that fermentation kinetics were affected by phytochemicals present [12].

The resulting acidity found in cannabis-infused kombucha consistently remained lower across the fermentation time, despite the fact that the LAB viability detected in those samples was higher (Figure 2b). In fact, lactic acid production was supposed to increase the total acidity [48]; however, it might have acted as a buffering capacity, thereby resisting significant pH changes [15]. As a result, cannabis-infused kombucha produced lower acidity and improved sensory appeal by reducing the harsh vinegar acidity taste.

A higher concentration of cannabis leaves added in kombucha increased the TSS content, particularly during the initial fermentation. As fermentation progressed, the TSS content significantly declined across all samples, specifically on day 3 of fermentation (Figure 3c). This could be due to yeast glycolysis, which breaks down sucrose [15]. The samples of cannabis-infused kombucha consistently exhibited a higher TSS content of 11.47–15.13 °Brix throughout fermentation, whereas kombucha tea was in the range of 11.10–12.30 °Brix. These results can be affected by the more water-soluble fractions of the substances present in the cannabis leaves, particularly the mineral content that easily dissolves in water (Table 1). Recent studies have confirmed that fermenting herbs in kombucha increases TSS levels due to the higher water-soluble compounds present in herb leaves [15,34]. This result suggests that cannabis leaves may provide additional fermentable substrates that benefit microbial growth. Furthermore, the solubilized fraction from cannabis leaves influenced the turbidity value in kombucha, which was consistent with the strong correlation between TSS and turbidity (r = 0.813; *p* < 0.01). Hence, the highest TSS (15.13 °Brix) was associated with the highest turbidity (11.68), and it was found in KACL50-D0. Turbidity was influenced by the fermentation time, as revealed by the decrease in turbidity from days 0 to 7 of fermentation (Figure 3d). Moreover, the turbidity also declined as the cannabis leaf concentration increased in a dose-dependent manner. This can be attributed to microbial growth converting complex solubilized substances into simpler compounds [45].

### 3.4. Effect of the Concentration of Cannabis Leaves and Fermentation Time on Total Phenolic Content and Antioxidant Properties of the Fermented Kombucha

The effects of different concentrations of cannabis leaves and fermentation times on TPC and DPPH antioxidant capacity were revealed (Figure 4). Initially, higher TPC and DPPH values were observed in KACL0-D0, peaking at 1.65 mg GAE mL^−1^ and 13.40 mg TE mL^−1^, respectively. Conversely, the infusion of cannabis leaves at higher concentrations (KACL30-D0 and KACL50-D0), was able to reduce the TPC and DPPH values by the ranges of 0.91–0.93 mg GAE mL^−1^ and 4.97–8.73 mg TE mL^−1^, respectively. Furthermore, a significant decline in TPC and DPPH values occurred as fermentation progressed, notably in the KACL50-D7 samples, as shown in Figure 4a,b. These samples exhibited a sharper reduction compared to KACL15-D7 and KACL30-D7 from day 0 to day 7, suggesting that a higher cannabis leaf concentration may accelerate the loss of TPC and DPPH antioxidant activity. These results were corroborated by the positive correlation between the TPC and DPPH (r = 0.719; *p* < 0.01). These findings implied that the change in antioxidant properties of kombucha were influenced by the polyphenols and antioxidant capacity presented in the plant substrates (Table 1) [2,9,15].

During the fermentation period, the decline in TPC and DPPH activity may be attributed to microbial and enzymatic activities involved in the degradation and transformation of polyphenol structures [51,52]. Specifically, oxygen radicals generated by microorganisms along with microbial enzymes (e.g., glucosidase, cellulase, and pectinase) could promote oxidation. The conversion of catechin gallate to non-gallate forms occurred, thereby lowering catechin content and DPPH radical scavenging activities [17,41]. Furthermore, the lower catechin content appeared to enhance microbial growth [53] and the large number of LAB and TYM counts were obtained in KACL30-D7 and KACL50-D7.

### 3.5. PCA and HCA Clustering of Kombucha Regarding the Quality Characteristics

Each kombucha treatment was clustered through PCA and HCA, based on similar quality and microbial characteristics during fermentation (Figure 5). PCA was used to analyze the correlations between kombucha treatments and fermentation variables to determine the most significant effects of treatments according to these factors. The PCA results indicated that the principal component (PC1) accounted for 57.83% and PC2 accounted for 24.91% of the total variance. Cluster 1 included four samples (KACL0-D0, KACL15-D0, KACL30-D0, KACL50-D0), which were positively correlated with pH, TSS, and turbidity, indicating no significant changes in quality and microbial characteristics during the initial fermentation. Cluster 2 was related to PC2 and included six samples (KACL0 and KACL15 from day 3 to day 7 of fermentation), which positively correlated with greater TPC and DPPH antioxidant capacity. These samples contained higher concentrations of assam green tea leaves, thus offering greater antioxidant properties. Cluster 3 included six samples (KACL30 and KACL50 from day 3 to day 7 of fermentation) and was related to larger TYM and LAB counts with less total acidity. Notably, KACL30-D7 was strongly correlated with the highest LAB and TYM counts due to higher cannabis leaf concentration along with longer fermentation time. Moreover, HCA highlighted the clustering of kombucha treatments to emphasize PCA results (Figure 5b). All clusters in HCA showed similar samples within each cluster, as revealed by the PCA results. In the HCA, KACL0-D7, and KACL30-D7 were separated by different clusters because KACL30-D7 was characterized by higher LAB and TYM counts and less harsh acidity, whereas KACL0-D7 showed opposite quality characteristics. These findings could be influenced by the concentration of cannabis leaves and fermentation time on day 7. Furthermore, they might be due to the symbiotic effect of microbial communities contained in kombucha [7]. The treatment with the highest LAB counts was used to select the final formulated kombucha, because LAB strains could improve the quality and modulate the changes in the microbial communities of kombucha [20]. Additionally, by day 7 samples were chosen due to the acidity content reaching 4–5 g/L as a parameter to ensure the microbial safety and desirable sensory characteristics of kombucha [54]. Therefore, KACL30-D7 was selected for metagenomic analysis to uncover the microbial communities and their effects on quality characteristics compared to KACL0-D7.

### 3.6. Effect of the Concentration of Cannabis Leaves on Microbial Communities and Diversity Changes in the Final Fermented Kombucha

#### 3.6.1. Fungal Communities and Diversity of Cannabis-Infused Kombucha: Effects on Fermentation and Quality Changes

The effect of cannabis leaf infusion on fungal community changes in fermented kombucha was revealed (Figure 6a). At the phylum level, cannabis-infused kombucha (KACL30-D7) consisted entirely of *Ascomycota* (100%), while kombucha tea (KACL0-D7) comprised *Ascomycota* (99.47%), with the remaining phyla being *Mucormycota* and *Basidiomycota*. At the genus level, *Dekkera* and *Zygosaccharomyces* were identified in different proportions in both the samples (Figure 6a). *Rhizopus*, *Hanseniaspora*, *Saccharomyces*, and *Candida* were only detected in KACL0-D7. The identified fungal phyla and genera have been documented as the primary yeast in fermented kombucha [4,7]. These results suggested that cannabis leaf-infused kombucha solely induced the growth of prominent yeasts due to the phytochemicals present in those leaves (Table 1); thus, it could reduce the fungal taxonomic diversity in KACL30-D7, which was consistent with previous findings [6].

The fungal alpha diversity suggested that KACL30-D7 showed Shannon (1.49) and Chao1 (19) indices, whereas KACL0-D7 exhibited Shannon (1.66) and Chao1 (52) indices. Additionally, the evenness values for KACL30-D7 and KACL0-D7 were 0.505 and 0.421, respectively. These findings suggest that KACL30-D7 had less fungal alpha diversity because of lower values in Shannon and Chao1, and it detected only the prominence of *Dekkera* and *Zygosaccharomyces*. KACL30-D7 showed the abundance of *Dekkera,* indicating a more equitable distribution as the dominant yeast, as supported by a greater evenness value. This might be induced by nitrogenous substances from the cannabis leaves (Table 1). Soluble proteins and other soluble phytonutrients, especially polyphenols, are essential for promoting the growth of microorganisms by maintaining fermentative metabolism and stimulating microbial activity [13,27]. This study suggested that the dominance of *Dekkera* yeast played a crucial role in kombucha fermentation, as it provides essential nutrients such as glucose, fructose, and ethanol [45]. Consequently, the availability of these nutrients could promote the continuous growth of other yeast strains in KACL0-D7 (Figure 6a). Thus, the fungal diversity in both samples highlighted the crucial roles of dominant yeast during fermentation and its impact on quality changes.

Metagenomic analysis revealed that *Dekkera* and *Zygosaccharomyces* were the core components of microbiota during kombucha fermentation. The proportion of *Dekkera* was more dominant in KACL30-D7 (85.13%) than in KACL0-D7 (72.80%), which can be attributed to *Dekkera*’s competitive advantage in nutrient-poor and stressful environmental conditions [5,11]. As the effect of *Dekkera* yeast’s dominance, the KACL30-D7 sample exhibited high TSS and pH, less acidity, and less harsh acetic taste. This could be due to the slower fermentation rate during ethanol oxidation caused by cannabis leaves and might be affected by the growth rate of *Dekkera* as a slow-growing yeast [11]. This finding is consistent with that of Wang et al. [12], who reported that fermentation occurred more slowly in the herb leaves of kombucha. Moreover, *Zygosaccharomyces* was identified in less abundance than *Dekkera* in both kombucha samples (Figure 6a). This could be due to the growth of *Zygosaccharomyces* being hindered by nutrient depletion and restricted oxygen supply because of the cellulose biofilm formation during longer fermentation times [5,11,42]. Specifically, KACL0-D7 showed a greater proportion of *Zygosaccharomyces* than KACL30-D7, as a result, it produced the opposite quality and microbial characteristics. In KACL0-D7, *Hanseniaspora*, *Saccharomyces*, and *Candida* provide nutrients to enhance the production of organic acids [7]; consequently, a lower pH with high acidity was achieved. By day 7 of fermentation, the decline in TPC and DPPH could be due to *Dekkera* and *Zygosaccharomyces* altering the structural polyphenols and degrading catechin content in both kombucha samples, as revealed in recent studies [1,52]. Additionally, the dominance of *Dekkera* and *Zygosaccharomyces* in KACL30-D7 may improve gut health, because both yeast strains can tolerate simulated gastrointestinal conditions and inhibit pathogenic microbes [4,5,6]. KACL0-D7 also identified *Hanseniaspora* and *Saccharomyces* as having the highest antioxidant activity and antibiotic resistance [4]. Hence, differences in the fungal community and diversity influenced the fermentation and quality changes in kombucha. Notably, the dominance of *Dekkera* in KACL30-D7 significantly enhanced overall quality by reducing harsh acidity and may offer gut health benefits.

#### 3.6.2. Bacterial Communities and Diversity of Cannabis-Infused Kombucha: Effects on Fermentation and Quality Changes

The cannabis leaf infusion led to diverse bacterial communities in KACL30-D7 (Figure 6b). This was emphasized by the presence of *Proteobacteria* (99.51%) and *Firmicutes* (0.42%) at phyla levels, with seven genera detected (Figure 6b). The KACL30-D7 and KACL0-D7 demonstrated *Proteobacteria* and *Komagataeibacter* as the prominent phylum and genus, respectively. Similar observations have been reported for kombucha across various regions, highlighting its primary role in acetic acid fermentation [7,49,55,56]. However, *Staphylococcus*, *Sphingomonas*, *Serratia*, and *Pseudomonas* were detected in KACL30-D7, likely originating from airborne contamination, environmental exposure, or incubator conditions, as fermentation occurs under aerobic conditions [6,19]. Previous studies have also reported the presence of these bacteria during kombucha fermentation. Despite these findings, the abundance of *Komagataeibacter* may inhibit pathogenic bacteria due to its production of acetic acid, which possesses antibacterial properties [2]. In KACL30-D7, a greater bacterial community was reinforced by *Weissella* as the LAB group [7,57] and other bacterial genera (Figure 6b). These findings suggest that the greater bacterial community in KACL30-D7 could be attributed to soluble phytonutrients from cannabis leaves (Table 1), which induce the growth of other bacterial genera, consistent with recent studies [44]. Thus, fermenting kombucha with different substrate matrices influenced bacterial profiles and diversity changes, as reported previously [19].

The alpha diversity indices of the bacteria in KACL30-D7 and KACL0-D7 were determined. KACL30-D7 exhibited a Chao1 index (18), Shannon index (0.587), and evenness value (0.203). Meanwhile, KACL0-D7 showed a Chao1 index (9), Shannon index (0.748), and evenness value (0.341). A greater Chao1 value in KACL30-D7 indicated the richness of the bacterial communities, as evidenced by the various bacterial genera (Figure 6b). The bacterial community richness in KACL30-D7 may be influenced by the dominance of *Dekkera*, which provides sufficient nutrients and optimal fermentation conditions [5,45]. Moreover, cannabis leaf contains phytonutrients and polyphenols (Table 1) that might induce the growth of several bacterial genera, specifically *Weissella*. Furthermore, a larger evenness value in KACL0-D7 suggested a more homogenous distribution of *Komagataeibacter* dominance. This could be due to its adaptability in utilizing nutrients under harsh environments during acetic acid fermentation, which is consistent with the findings of Wu et al. [5]. In KACL0-D7, the dominance of *Komagataeibacter*, along with the presence of *Saccharomyces* and *Zygosaccharomyces*, may impede the growth of other bacteria by creating high levels of acid and ethanol [42]. Hence, bacterial alpha diversity has a significant effect on kombucha fermentation and quality changes.

Although *Komagataeibacter* was dominant, its proportion and the presence of other genera affected a difference in quality and microbial characteristics between KACL0-D7 and KACL30-D7. KACL30-D7 demonstrated a higher pH with less harsh acidity (Figure 3). This could be because the proportion of *Komagataibacter* was slightly lower in KACL30-D7 than in KACL0-D7, consistent with recent findings [58]. Furthermore, *Weissella* could moderate acidity by softening acetic acid through lactic acid production, which is consistent with the results of Wang et al. [20]. As a heterofermentative strain, *Weissella* may facilitate a buffering effect through lactic acid and carbon dioxide production [19,43]. Therefore, a higher pH was obtained in KACL30-D7. Furthermore, the dominance of *Komagataibacter* led to a decline in TPC and DPPH during final fermentation (Figure 4), by mediating polyphenol biotransformation and degradation, as revealed in previous studies (1). The kombucha functionalities in KACL30-D7 may be obtained from *Komagataeibacter* and *Weissella* to promote gut health benefits [3,58]. *Komagataeibacter* has been reported as a novel probiotic candidate due to its survival in simulated digestive systems with a high glucose conversion rate [3]. *Weissella* has demonstrated high survivability in simulated gastrointestinal environments and has exhibited the production of short-chain fatty acids, as well as antibiotic sensitivity and antimicrobial activity [59,60]. In KACL30-D7, *Weissella* was present in low abundance, likely due to competition with the dominant *Dekkera* and *Komagataeibacter* during glycolysis and pyruvate metabolism [42,57]. These findings suggested that the LAB strains in KACL30-D7 can modulate kombucha fermentation, improve quality characteristics, and promote digestive health.

### 3.7. Effect of Microbiota Communities on Volatile Compounds Changes in the Fermented Kombucha Infused with Cannabis Leaves

The identified volatile compounds found in kombucha samples (KACL0 and KACL30) during day 0 and day 7 of fermentation are shown in Figure 7. A total of 64 volatile compounds were identified in the samples, including alcohols, acids, esters, hydrocarbons, aldehydes, ketones, and phenols. The results indicated that kombucha samples exhibited an increase in volatile compounds from day 0 to day 7 of fermentation. Specifically, KACL30 showed an increase in volatile compounds from 32 to 39, while KACL0 exhibited a change from 29 to 39 volatile compounds (Figure 7). These findings suggest that fermentation modulates volatile compound profiles through biotransformation performed by microbial communities and their enzymes. This phenomenon leads to the accumulation, degradation, or conversion of volatile compounds, which was consistent with previous reported observations [6,19].

During initial fermentation (day 0), both kombucha samples exhibited major volatile compounds, including ethanol, cyclohexene, 1-methyl-4-1-methyl ethylidene, acetic acid, and ethyl acetate. A Venn diagram revealed significant differences in volatile compounds between KACL0-D0 (8 unique compounds) and KACL30-D0 (11 unique compounds), while 21 compounds were common to both, though in varying proportions (Figure 7a). Notably, ethanol content was lower in KACL30-D0 (13.81%) when cannabis leaves were infused compared to the control kombucha tea (25.47%). This reduction may be attributed to differences in ethanol-producing yeast strains [9], as the control kombucha contained various strains, including *Zygosaccharomyces*, *Saccharomyces*, *Talaromyces*, and *Hanseniaspora* (Figure 6a). Similarly, acetic acid content was lower in KACL30 compared to the control. Acetic acid in kombucha is primarily produced by acetic acid bacteria from the SCOBY kombucha culture. As shown in Figure 6b, KACL0 was only identified *Komagataibacter*, while KACL30 exhibited greater bacterial diversity, which could affect acetic acid production. Interestingly, on day 0, KACL30-D0 exclusively contained eugenol and benzaldehyde, imparting fruity, herbal, and clove-like aromas, whereas the control kombucha had higher levels of 2-propyl-1-pentanol and linalool, contributing to a more pronounced alcoholic and floral aroma [19]. These findings suggest that raw material influences the volatile profile composition, which was found during early fermentation. Similar studies have reported the same phenomenon [6,19].

By day 7 of fermentation, the Venn diagram revealed significant differences in eight volatile compounds between the KACL0-D7 and KACL30-D7 samples, whereas 31 compounds were common to both samples, albeit in different proportions (Figure 7b). KACL30-D7 showed lower levels of ethanol (9.79%), carbon dioxide (5.07%), and acetic acid (25.69%). This could be influenced by a higher abundance of *Dekkera* as the effect of cannabis leaf infusion, which is consistent with previous studies [9,19]. The slow growth rate of *Dekkera* leads to a slower fermentation rate of sugar consumption [11]. Similar results were obtained when fermenting herb leaves of kombucha [12]. In this study, the ethanol content was decreased during the final fermentation, which could be further converted into acetic acid or undergo esterification to produce esters [6]. Moreover, the proportion of *Komagateibacter* and the abundance of LAB counts found in KACL30-D7 alleviated harsh vinegar by reducing acetic acid and agreed with recent findings [20,58]. Conversely, KACL0-D7 showed higher levels of ethanol, acetic acid, and carbon dioxide. This might be attributed to the presence of *Saccharomyces*, *Hanseniospora*, and a high abundance of *Zygosaccharomyces* (Figure 6a). These yeasts are commonly involved in faster ethanol production [11,61]. Consequently, a higher content of acetic acid was produced, which correlated with the dominance of *Komagateibacter* in KACL0-D7, as revealed previously [58]. Thus, infusing cannabis leaves into kombucha improved flavor quality and moderated vinegar acidity.

On day 7, KACL30-D7 exhibited the formation of new volatile compounds, including phenol and ester groups, which emerged only at the end of fermentation. KACL30-D7 exhibited higher levels of ester and phenol compounds, with eugenol and phenol, 2,3,5,6-tetramethyl were exclusively identified (Figure 6). In KACL30-D7, the abundance of ester and phenol compounds can be influenced by the dominance of *Dekkera*, which possesses high esterase activity to release fruit and floral-like aromas. Notably, in KACL30-D7, acetic acid oxo-methyl ester and 9-octadecenoic acid (Z)-methyl ester were detected at higher concentrations. The abundance of *Dekkera* yeast and LAB counts (Figure 2b) might promote the production of desired metabolites, specifically ester compounds, consistent with recent findings. Phenolic compounds are produced through reactions catalyzed by β-glucosidase enzymes and involve catechin methylation. In KACL30-D7, eugenol was identified as exclusively imparting phenolic and clove flavors. This may be attributed to the metabolic activities of β-glucosidase and esterase in the *Dekkera* yeast. KACL0-D7 contained significant amounts of dodecanoic acid and exclusively tetradecanoic acid (Figure 6), imparting rancid and fatty odors. These compounds can be attributed to the metabolism of *Zygosaccharomyces* and *Komagataibacter* [5,42]. The compounds identified exclusively in KACL0-T4, including 1-butanol-3-methyl and 1-hexanol-2-ethyl, are responsible for the activity of *Talaromyces*, *Saccharomyces*, and *Hanseniospora* strains. Hence, KACL30-D7 has a fruity aroma with a less vinegar-like smell, whereas KACL0-D7 has harsh vinegar and alcoholic odors. These findings suggest that infusing cannabis leaves influenced the changes in volatile compounds due to greater bacterial richness and fungal evenness diversity with the prominence of *Dekkera.*

## 4. Conclusions

When considering the quality characteristics and the microbial benefits of the resulting kombucha, the appropriate cannabis infusion was 30% of the concentration. This sample showed the highest growth of LAB and total yeast, presence of *Weissella* bacteria, and abundance of *Dekkera* and *Komagataeibacter*, by day 7 of fermentation. *Komagataeibacter*, *Dekkera*, and *Zygosaccharomyces* were the core microbiota during kombucha fermentation and potentially benefiting gut health. These strains altered the quality of cannabis-infused kombucha and resulted in lessening ethanol and acetic acid content, containing higher ester compounds, and lowering vinegar acidity. The polyphenol content and DPPH antioxidant capacity were lower than those of the control kombucha tea. However, the cannabis-infused kombucha had greater taste and smell, while remaining safe for consumption conforming to the Thai regulatory threshold which is less than 1.41 mg THC and 1.60 mg CBD per 1 serving size, and consequently, showed its potential symbiotic and postbiotic effects.

## Figures and Tables

**Figure 1 foods-14-00942-f001:**
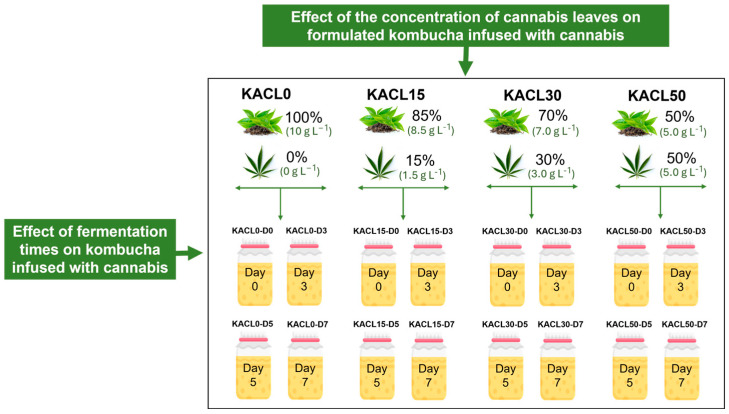
The obtained treatment of kombucha at different cannabis concentrations and fermentation times.

**Figure 2 foods-14-00942-f002:**
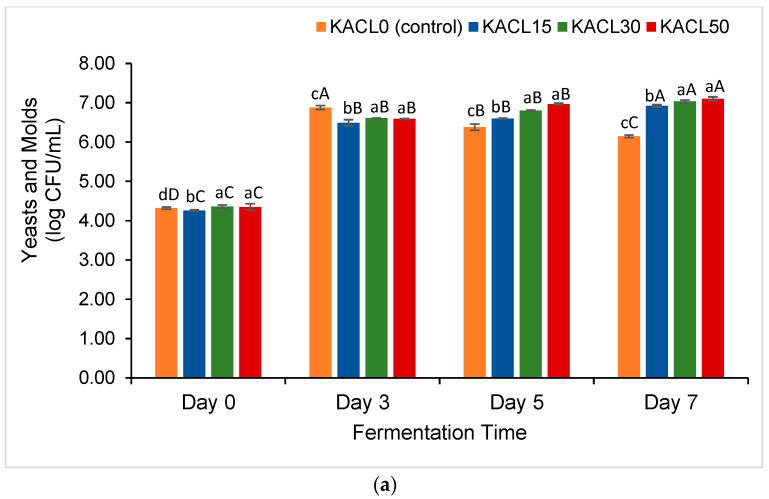
Effect of infused cannabis leaf concentration and fermentation times on microbial growth across kombucha samples (**a**) yeast and molds and (**b**) lactic acid bacteria. Different letters (a–d) within the same fermentation day indicate significant differences in microbial growth between samples of kombucha with different concentrations of cannabis leaf (*p* < 0.05). Different letters (A–D) within the same color of kombucha samples indicate significant differences in microbial growth across fermentation times (*p* < 0.05).

**Figure 3 foods-14-00942-f003:**
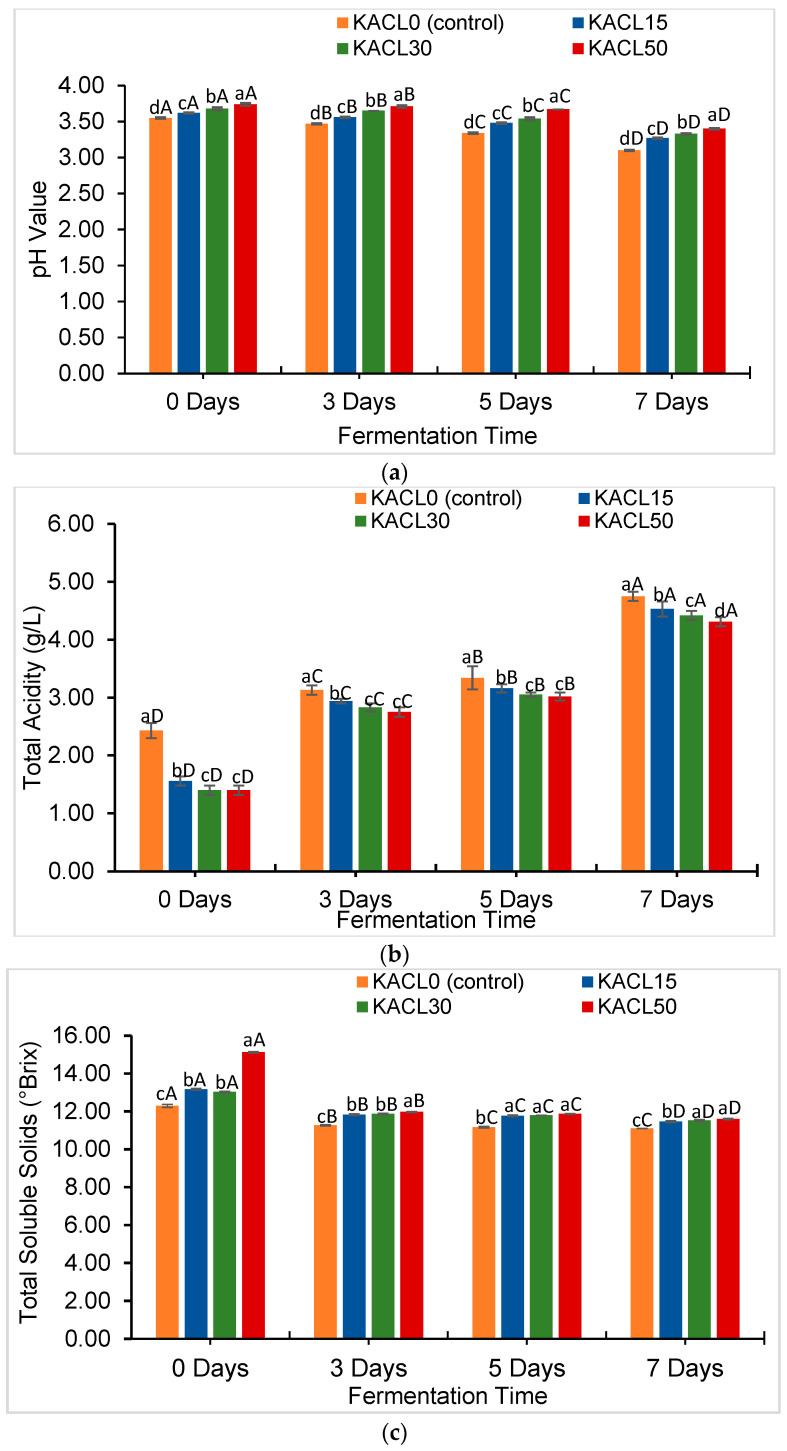
Effect of infused cannabis leaf concentration and fermentation times on changes in quality characteristics across kombucha samples including (**a**) pH, (**b**) total acidity, (**c**) total soluble solids, and (**d**) turbidity. Different letters (a–d) within the same fermentation day indicate significant differences in quality characteristics between samples of kombucha with different concentrations of cannabis leaf (*p* < 0.05). Different letters (A–D) within the same color of kombucha sample indicate significant differences in quality characteristics across fermentation times (*p* < 0.05).

**Figure 4 foods-14-00942-f004:**
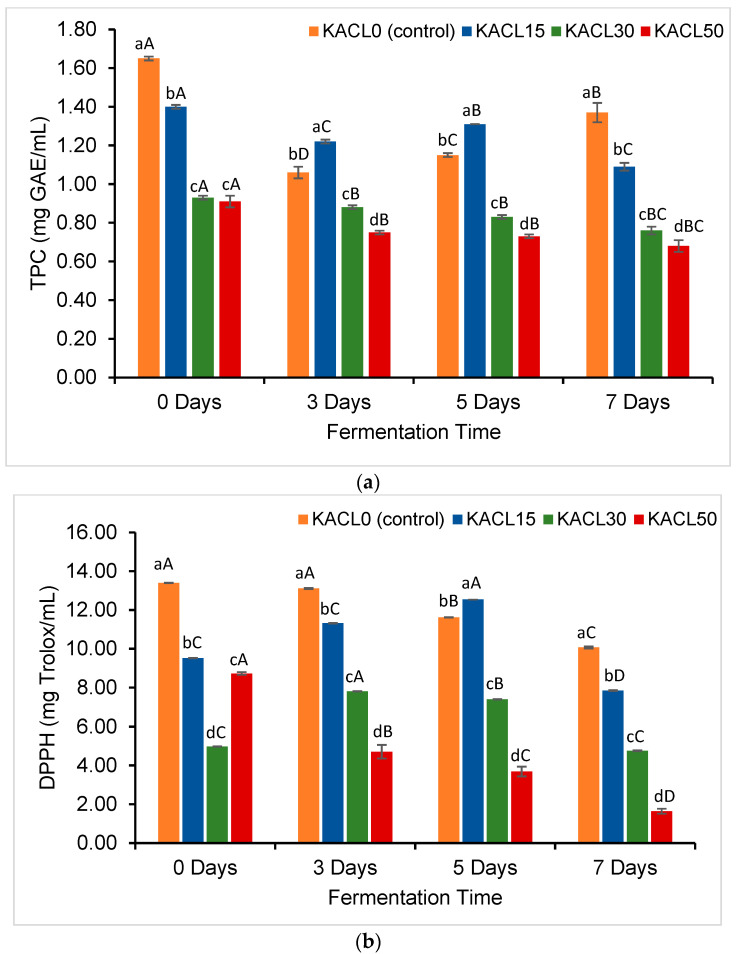
Effect of infused cannabis leaf concentration and fermentation times on changes in antioxidant properties across kombucha samples including (**a**) TPC and (**b**) DPPH. Different letters (a–d) within the same fermentation day indicate significant differences in antioxidant properties between samples of kombucha with different concentrations of cannabis leaf (*p* < 0.05). Different letters (A–D) within the same color of kombucha sample indicate significant differences in antioxidant properties across fermentation times (*p* < 0.05).

**Figure 5 foods-14-00942-f005:**
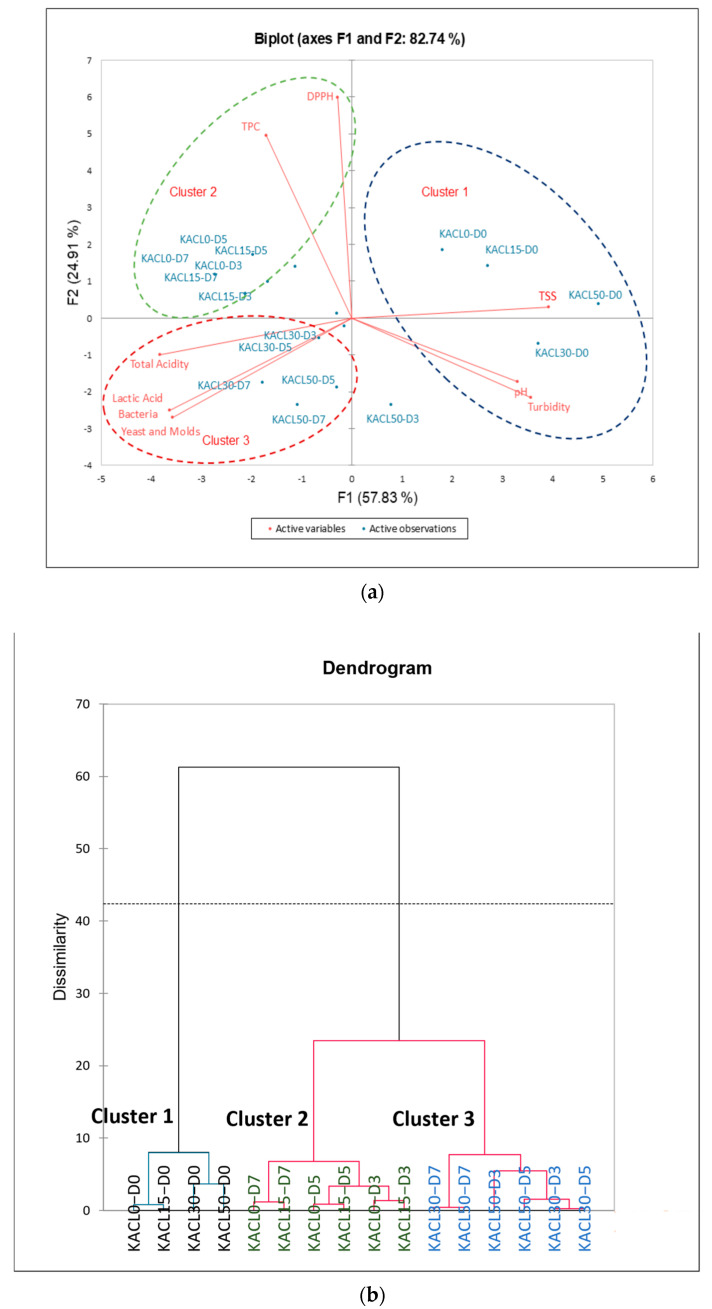
Clustering of kombucha treatments based on quality characteristics changes and microbial growth (**a**) PCA (**b**) HCA.

**Figure 6 foods-14-00942-f006:**
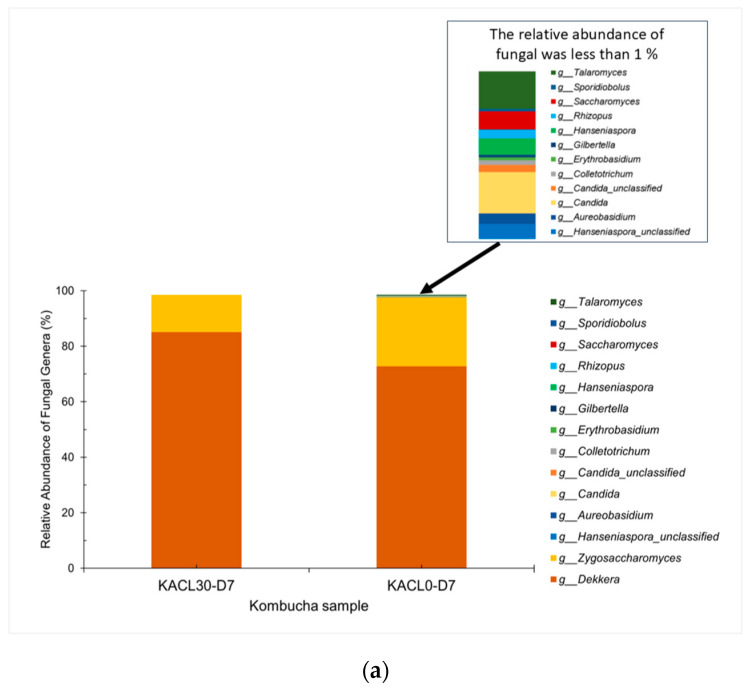
Effect of cannabis leaf infusion on microbiota at genus levels in the final of fermented kombucha (**a**) fungal community (**b**) bacterial community.

**Figure 7 foods-14-00942-f007:**
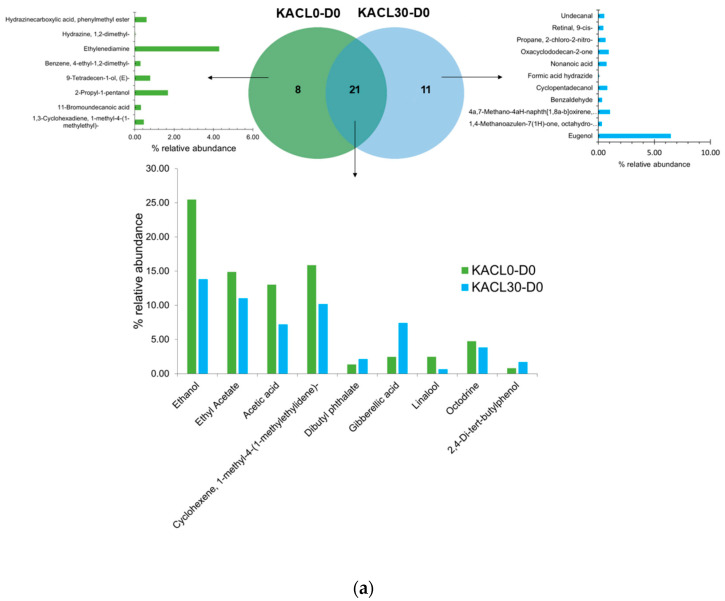
The changes in volatile compounds in the fermented kombucha infused with cannabis leaves (**a**) Initial fermentation (day 0) (**b**) Final fermentation (day 7).

**Table 1 foods-14-00942-t001:** Antioxidant properties and proximate composition of plant materials used for kombucha fermentation.

Parameters	Assam Green Tea Leaves	Cannabis Leaves
TPC (mg GAE/mL extract)	1.08 ± 0.01 ^a^	0.10 ± 0.00 ^b^
DPPH (mg Trolox/mL extract)	21.28 ± 1.06 ^a^	0.63 ± 0.01 ^b^
Ash (g/100 g)	6.23 ± 0.10 ^b^	14.79 ± 0.20 ^a^
Moisture (g/100 g)	2.98 ± 0.04 ^b^	9.96 ± 0.11 ^a^
Crude fat (g/100 g)	0.45 ± 0.02 ^b^	4.09 ± 0.12 ^a^
Crude protein (g/100 g)	21.68 ± 0.68 ^b^	23.10 ± 0.17 ^a^
Crude fiber (g/100 g)	21.27 ± 0.58 ^a^	16.92 ± 1.20 ^b^
Carbohydrates (g/100 g)	47.39 ± 0.68 ^a^	31.14 ± 1.57 ^b^

^(a,b)^ mean ± standard deviations in the same row indicated significant differences (*p* < 0.05). Values of proximate composition are presented on a wet weight basis.

## Data Availability

Data are contained within the article.

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
