# Peer review of "Thai Cannabis sativa Leaves as a Functional Ingredient for Quality Improvement and Lactic Acid Bacterial Growth Enhancement in Kombucha"

_foods, 2025, doi:10.3390/foods14060942_

Round 1
Reviewer 1 Report
Comments and Suggestions for Authors
The content of the article is scientifically interesting and the product (kombucha), being a fermented drink with bioactive properties, is certainly commercially attractive.
Below I add comments to improve the clarity of some important points covered in the manuscript.
1) In some parts of the article, as well as in the Abstract, it is stated that the nitrogen content of cannabis leaves probably stimulated the growth of LAB. However, care must be taken as to the nitrogen content in the extract. Table 1 shows the nitrogen content of the leaves and not the original extract that was fermented. Review this type of statement throughout the article. The effect on growth must have been influenced more by other components of cannabis than by its nitrogen content.
- Abstract: “The high protein content (23.10 %) of Cannabis leaves served as a nitrogen source for promoting LAB growth which resulted in the higher number of LAB in the treatment with can- nabis leaves”
-Line 293: These findings suggested the synergistic effect of cannabis leaves with microbial growth by providing phytonutrients, mainly nitrogen sources…
- Line 530: “ These findings suggest that the greater bacterial community in KACL30-D7 could be attributed to soluble phytonutrients from can…” Yes! I agree
- Line 542: Moreover, greater nitrogen sources from cannabis leaves (Table 1), may induce the growth of several bacterial genera, specifically Weissella
Why “mainly nitrogen?? How could these results be associated with the nitrogen source? Green tea and cannabis leaves have similar N content. Don’t other compounds present in the cannabis extract be associated with the better growth of LAB?
2) Pure cannabis extract stimulated LAB growth in rats, but would the fermented extract, i.e. the final drink, do the same? What happens to THC and CBD after fermentation? Please include a discussion of this in the text, here:
- Line 81: Furthermore, pure extracts of cannabis leaves have been shown to enhance the growth of Bifidobacterium and Lactobacillus, in mice due to cannabinoid compounds, thereby benefiting gut health [21,26].
3) Line 277 – 278: Yes! I agree the composition of plant materials may provide carbon and nitrogen sources, but if it is “… beneficial for microbial growth and affect the overall quality characteristics during kombucha fermentation” it is not clear. In the case of carbon sources, the addition of sucrose is necessary to perform the fermentation. I suggest review this statement. Nitrogen content of the plants/leaves is not the content of nitrogen in the extract that will be fermented. Was the content of N measured in the tea infusion/extract?
4) Is it correct? 500 and 1000 rpm?
Line 239-238: 500 rpm for 10 minutes, followed by continuous stirring at 1000 rpm and heating at 70°C for 30 minutes
5) The title of table 1 need to be correct: “Table 1. This is a table. Tables should be placed in the main text near to the first time they are cited”.
6) Figura 6b: How the presence of other bacterial genera like Staphylococus, Sphingomonas, Serratia, Pseudomonas affect the kombucha? These genera commonly appear in kombucha? Please comment.
7) Line 296: When you say “the phytochemical cannabis extract enhanced the growth of Lactobacillus strains and inhibited pathogenic species..” it is meaning the presence of LAB and probably its bacteriocins or the decrease of pH inhibits the pathogenic species, and not due to the phytochemicals present in cannabis extract. Review this sentence. Because the effect of the cannabis extract cannot be associate in this study with the inhibition of pathogenic species.
8) Line 233: Volatile compound analysis was conducted with the sample selected from 2.4 according to the method of Wang et al. [38].
Were the volatile compounds of the initial extract with green tea and cannabis leaves analyzed to compare the volatile compounds produced during the kombucha fermentation? Include in the discussion ( Figure 7 ). Maybe many of those 31 compounds common to both samples were already present in the original extract.
Author Response
Response to Reviewer
Reviewer#1
Title: Thai Cannabis sativa Leaves as Functional Ingredient for Quality Improvement and Lactic Acid Bacterial Growth Enhancement in Kombucha
The content of the article is scientifically interesting and the product (kombucha), being a fermented drink with bioactive properties, is certainly commercially attractive.
Below I add comments to improve the clarity of some important points covered in the manuscript.
Response: Thank you very much for your kind suggestions and for spending your valuable time revising our manuscript. All suggestions and comments help improve the merit and quality of our manuscript. If there are still any further corrections, please do not hesitate to let us know. Thank you very much again.
1) In some parts of the article, as well as in the Abstract, it is stated that the nitrogen content of cannabis leaves probably stimulated the growth of LAB. However, care must be taken as to the nitrogen content in the extract. Table 1 shows the nitrogen content of the leaves and not the original extract that was fermented. Review this type of statement throughout the article. The effect on growth must have been influenced more by other components of cannabis than by its nitrogen content.
Response: Thank you very much for your kind suggestions. It is true that other compounds that might arise during fermentation account for the LAB growth enhancement. According to the report of Yang et al. (2023), the relationship between phytochemicals in plant leaves and LAB growth was revealed. They reported that phytochemical (polyphenols) is the substrate for LAB and can inhibit the growth of pathogenic bacteria. Hence, the growth of LAB was promoted. We’ve already added the information in the Introduction part (Line 48-53).
Moreover, we’ve already read through the manuscript to revise the sentences for a better understanding as shown in the red alphabet. Thank you very much again.
- Abstract: “The high protein content (23.10 %) of Cannabis leaves served as a nitrogen source for promoting LAB growth which resulted in the higher number of LAB in the treatment with cannabis leaves”
Response: Thank you very much for your kind suggestions. We’ve already corrected the sentence as shown in the abstract part (Line 19-21). Thank you very much again.
-Line 293: These findings suggested the synergistic effect of cannabis leaves with microbial growth by providing phytonutrients, mainly nitrogen sources…
Response: Thank you very much for your kind suggestions. We’ve already corrected the sentence as shown in the Results part (Line 282-284 and 298-304). Thank you very much again.
- Line 530: “ These findings suggest that the greater bacterial community in KACL30-D7 could be attributed to soluble phytonutrients from can…” Yes! I agree
Response: Thank you very much for your kind suggestions. This sentence remained the same while other sentences as stated above were revised.
- Line 542: Moreover, greater nitrogen sources from cannabis leaves (Table 1), may induce the growth of several bacterial genera, specifically Weissella.
Response: Thank you very much for your kind suggestions. We’ve already corrected the sentences as shown in the Results part (Line 560-561). Thank you very much again.
Why “mainly nitrogen?? How could these results be associated with the nitrogen source? Green tea and cannabis leaves have similar N content. Don’t other compounds present in the cannabis extract be associated with the better growth of LAB?
Response: Thank you for your insightful question and the valuable suggestions. We’ve already reviewed more and found that the enhanced LAB growth observed in our study may not be attributed solely to nitrogen sources, although nitrogen availability can support LAB growth optimally. Both green tea and cannabis leaves contain comparable nitrogen levels (with statistically significant differences in protein content). However, cannabis leaves also contain cannabinoids, which previous study reported as antimicrobial agent and may influence microbial dynamics during kombucha fermentation. In our study, we analyzed CBD and THC content in dried cannabis leaves (results not shown) which is our raw material, but the direct impact on LAB growth still needs further investigation. Hence, we revised the misunderstood information as shown in the red alphabet in the Results part. Thank you very much again.
2) Pure cannabis extract stimulated LAB growth in rats, but would the fermented extract, i.e. the final drink, do the same? What happens to THC and CBD after fermentation? Please include a discussion of this in the text, here:
- Line 81: Furthermore, pure extracts of cannabis leaves have been shown to enhance the growth of Bifidobacterium and Lactobacillus, in mice due to cannabinoid compounds, thereby benefiting gut health [21,26].
Response: Thank you for your insightful question. In this study, we quantified CBD and THC content in dried cannabis leaves before fermentation to assess their presence in the raw material. However, due to confidentiality restrictions, we are unable to disclose the full dataset in this manuscript. The available data indicates that CBD was measured at 0.353 mg/g and THC at 8.054 mg/g in the dried leaves. Notably, the CBD and THC content added in the kombucha drinks was not exceed the limit of use by Thai regulation. Hence, we can use this leaf as food. Since hydrolysis occurs during, the change of CBD to THC might be resulted. We’ve reached out more information and the sentences (Line 85-93) were revised. We appreciate your valuable feedback in enhancing the scientific rigor of our manuscript. Thank you very much again.
3) Line 277 – 278: Yes! I agree the composition of plant materials may provide carbon and nitrogen sources, but if it is “… beneficial for microbial growth and affect the overall quality characteristics during kombucha fermentation” it is not clear. In the case of carbon sources, the addition of sucrose is necessary to perform the fermentation. I suggest review this statement. Nitrogen content of the plants/leaves is not the content of nitrogen in the extract that will be fermented. Was the content of N measured in the tea infusion/extract?
Response: Thank you for your insightful comment. We’ve already corrected and rewritten the sentences to reveal a better understanding of LAB growth enhancement as shown in the text (Line 283-288). Thank you very much again.
4) Is it correct? 500 and 1000 rpm?
Line 239-238: 500 rpm for 10 minutes, followed by continuous stirring at 1000 rpm and heating at 70°C for 30 minutes
Response: Thank you for your valuable comment. Yes, the parameters are correct. In our study, agitation was performed gradually, starting at 500 rpm for 10 minutes, followed by continuous stirring at 1000 rpm while heating at 70°C for 30 minutes to entrap volatile compounds. We have ensured that this description is accurately presented in the manuscript. Thank you very much again.
5) The title of table 1 need to be correct: “Table 1. This is a table. Tables should be placed in the main text near to the first time they are cited”.
Response: Thank you for your suggestion. We have added a label to Table 1: "Antioxidant properties and phytochemicals composition of plant materials used for kombucha fermentation" and moved to place near the first citation main text already (Line 279-282). Thank you very much again.
6) Figura 6b: How the presence of other bacterial genera like Staphylococus, Sphingomonas, Serratia, Pseudomonas affect the kombucha? These genera commonly appear in kombucha? Please comment.
Response: Thank you for your valuable questions and comments. Due to the technique that we used to analyze the genera of microbial community, all possible microbes were detected and reported. The detection of the mentioned genera might be due to the death cells of the microbes after sterilizing the cannabis and Assam green tea extracts at 121 °C for 15 minutes. Moreover, the relative abundance remained very low (less than 1%) revealed that the mentioned genera are not core members of the kombucha microbiome. Their detection highlights the importance of strict microbial quality control in fermentation. We have incorporated this discussion into the manuscript to address this point. Thank you again for your critical question and valuable feedback.
7) Line 296: When you say “the phytochemical cannabis extract enhanced the growth of Lactobacillus strains and inhibited pathogenic species..” it is meaning the presence of LAB and probably its bacteriocins or the decrease of pH inhibits the pathogenic species, and not due to the phytochemicals present in cannabis extract. Review this sentence. Because the effect of the cannabis extract cannot be associate in this study with the inhibition of pathogenic species.
Response: Thank you for your valuable suggestions. We acknowledge that our study did not directly assess whether the phytochemicals in cannabis extract were responsible for inhibiting pathogenic species. Instead, the observed inhibition is more likely attributed to the presence of LAB and probably its bacteriocins or the decrease of pH inhibits the pathogenic species, and not due to the phytochemicals present in cannabis extract. To ensure accuracy, we have revised the sentence as shown in the red alphabet in the text (Line 302-309). We appreciate your valuable feedback, which has helped improve the clarity and scientific rigor of our manuscript. Thank you very much again for your thoughtful suggestions.
8) Line 233: Volatile compound analysis was conducted with the sample selected from 2.4 according to the method of Wang et al. [38].
Were the volatile compounds of the initial extract with green tea and cannabis leaves analyzed to compare the volatile compounds produced during the kombucha fermentation? Include in the discussion ( Figure 7). Maybe many of those 31 compounds common to both samples were already present in the original extract.
Response: Thank you for your insightful questions and valuable suggestions. In our study, we analyzed the volatile compounds in kombucha at day 0, immediately after introducing the kombucha culture to the green tea and cannabis extract mixture, rather than analyzing each extract separately. This could be due to our objective was to investigate the changes of volatile compound among kombucha tea and with infusing Cannabis leaves during fermentation and to compare the compositions at day 0 and day 7. Our findings indicate that fermentation significantly modified the volatile composition, leading to the formation of new compounds and alterations in the relative abundance of pre-existing ones. Regarding the 31 volatile compounds detected after fermentation, not all were originally present in the extract. Several compounds, such as acetic acid oxo-methyl ester, n-decanoic acid, octanoic acid, octanoic acid ethyl ester, phenol 2,3,5,6-tetramethyl-, and phenol 2-methoxy-4-propyl, resulted from ethanol conversion and esterification during fermentation. Furthermore, differences in these compounds between samples were mainly in their relative abundances. Specifically, the kombucha with cannabis leaves infusion had lower levels of acetic acid, carbon dioxide, and ethanol. Notably, the KACL30-D7 sample exhibited a more pronounced fruity aroma due to a higher relative percentage of ester compounds (acetic acid oxo-methyl ester and 9-octadecenoic acid (Z)-methyl ester).
To provide a more comprehensive perspective, we have included data on the volatile compounds detected at day 0 in both cannabis-infused kombucha and control kombucha tea. The number of volatile compounds was increased as shown in Figure 7a and 7b. These additions were rewritten in red alphabet and the Figure was revised.
We appreciate your valuable comments and suggestions which truly help us improve the quality of our manuscript. Thank you very much again.

Reviewer 2 Report
Comments and Suggestions for Authors
Thai Cannabis sativa Leaves as Functional Ingredient for Quality Improvement and Lactic Acid Bacterial Growth Enhancement in Kombucha
Authored by: Qurrata A’yuni, Kevin Mok, Massalin Nakphaichit, Kamolwan Jangchud and Tantawan Pirak
This study is focused on Cannabis sativa leaves as possible novel substrate for infusing in kombucha fermentation. The effect of cannabis leaf concentration and fermentation time on lactic acid bacteria growth, changes in quality characteristics, volatile compounds and microbial communities have been demonstrated and thoroughly analyzed.
This paper is well written with clearly justified and consistent descriptions. The authors have explained in detail why they chose cannabis leaves as a supplement for kombucha fermentation. Also, the applied statistical analyses are appropriate for revealing key information in this investigation.
For better understanding, "assam" and "Assam" should be standardized.
There are some places in the text that abbreviations can be used instead of full names, for instance delta-9-tetrahydrocannabinol at line 75 and 2,2-diphenyl-1-picrylhydrazyl at line 124.
For Table 1, insert a label.
Elucidate the meaning and complement the following sentences, due to the contradiction in their meaning: "When cannabis leaves were infused into kombucha, the TSS content was increased. As fermentation time increased, the TSS content significantly decreased across all samples (Figure 3c) and declined on day 3 of fermentation."
For clarity give additional explanations for the ranges at line 402 (which kombucha samples are compared) and connect the explanations with the figure.
In Figure 5a Brix could be changed with TSS.
At line 597 Figure 6 should be Figure 7.
Also, the authors should include the explanation of possible negative effects of infusion of Cannabis sativa leaves in kombucha fermentation for the consumers health.
There is no real-world testing experiment with the distinguished kombucha sample CACL30-D7.
Author Response
Response to Reviewer
Reviewer#2
Title: Thai Cannabis sativa Leaves as Functional Ingredient for Quality Improvement and Lactic Acid Bacterial Growth Enhancement in Kombucha
- This study is focused on Cannabis sativaleaves as possible novel substrate for infusing in kombucha fermentation. The effect of cannabis leaf concentration and fermentation time on lactic acid bacteria growth, changes in quality characteristics, volatile compounds and microbial communities have been demonstrated and thoroughly analyzed.
Response: Thank you for spending your valuable time reviewing our manuscript. Your comments and suggestions help improve the merit of our manuscript. Thank you very much again.
- This paper is well written with clearly justified and consistent descriptions. The authors have explained in detail why they chose cannabis leaves as a supplement for kombucha fermentation. Also, the applied statistical analyses are appropriate for revealing key information in this investigation.
Response: Thank you for your positive feedback and for taking the time to review our manuscript. We sincerely appreciate your recognition of our justification for using cannabis leaves in kombucha fermentation and the appropriateness of our statistical analyses. Your comments are kind to us. Thank you very much again.
- For better understanding, "assam" and "Assam" should be standardized.
Response: Thank you for your valuable suggestion. We have standardized the term to “assam” throughout the manuscript to ensure consistency. We appreciate your feedback in improving the clarity of our work. Thank you very much again.
- There are some places in the text that abbreviations can be used instead of full names, for instance delta-9-tetrahydrocannabinol at line 75 and 2,2-diphenyl-1-picrylhydrazyl at line 124.
Response: Thank you for your insightful suggestion. We have now introduced abbreviations for delta-9-tetrahydrocannabinol (THC) and 2,2-diphenyl-1-picrylhydrazyl (DPPH) at their first mention and used the abbreviations consistently throughout the manuscript as shown as the red alphabet in the text. This suggestion helps improve the quality of our manuscript. Thank you very much again.
- For Table 1, insert a label.
Response: Thank you for your suggestion. We have added a label to Table 1 titled "Antioxidant properties and phytochemical composition of plant materials used for kombucha fermentation.” as shown in Results part (Line 279). Thank you very much again.
- Elucidate the meaning and complement the following sentences, due to the contradiction in their meaning: "When cannabis leaves were infused into kombucha, the TSS content was increased. As fermentation time increased, the TSS content significantly decreased across all samples (Figure 3c) and declined on day 3 of fermentation."
Response: Thank you for your valuable comment. We have clarified the explanation to resolve the contradiction. The initial increase in TSS was due to the release of soluble compounds from cannabis leaves upon infusion. However, as fermentation progressed, microbial activity led to a significant decline in TSS, particularly by day 3, as sugars were metabolized. We have revised the text accordingly for clarity:
"A higher concentration of cannabis leaves in kombucha increased the TSS content, particularly during the initial fermentation. As fermentation progressed, the TSS content significantly declined across all samples, specifically on day 3 of fermentation (Figure 3c)." (Line 387-390).
Thank you very much again for your kind suggestions.
- For clarity give additional explanations for the ranges at line 402 (which kombucha samples are compared) and connect the explanations with the figure.
Response: Thank you for your valuable comment. We have revised the manuscript to provide additional explanations regarding the compared kombucha samples and have explicitly linked the discussion to Figures 4a and 4b.
Our findings indicate that a significant decline in TPC and DPPH antioxidant activity was observed as fermentation progressed, particularly in the KACL50-D7 samples. This sample exhibited a sharper reduction in comparison to KACL15-D7 and KACL30-D7 from day 0 to day 7. The revised text now clarifies these comparisons and provides a direct reference to Figures 4a and 4b, which visually elucidates this trend as shown in the text in Results part (Line 414-419).
We truly appreciate your insightful feedback, as it has helped improve the clarity and scientific merit of our discussion. Thank you very much again.
- In Figure 5a Brix could be changed with TSS.
Response: Thank you for your suggestion. We have replaced "Brix" with "TSS" in Figure 5a to improve clarity. We appreciate your valuable input in enhancing our manuscript quality. Thanks once again for your kind support.
- At line 597 Figure 6 should be Figure 7.
Response: Thank you very much for your kind suggestion and help us to improve the merit and quality of our manuscript. We’ve already corrected the number of Figure in the text (Line 603). Thank you very much again.
- Also, the authors should include the explanation of possible negative effects of infusion of Cannabis sativaleaves in kombucha fermentation for the consumers health.
Response: Thank you very much for your valuable comment. In this study, the concentration of Cannabis sativa leaves used in kombucha fermentation complies with Thai FDA regulations, ensuring that THC and CBD levels remain within the permitted limits, which is well below the regulatory threshold of 1.41 mg THC and 1.60 mg CBD per serving size set by Thai authorities. Furthermore, several commercial beverages in Thailand also incorporate cannabinoids while maintaining safety standards for consumption. Moreover, the report of Jangchud et al. (2024) highlighted that cannabis consumption may influence physiological parameters, such as blood pressure, heart rate, body temperature, memory, and cognitive function, reinforcing the importance of proper dosage control and labeling. Therefore, we suggest that exceeding the regulated THC and CBD limits could pose potential risks and must be carefully considered in product development. To clarify this point, we’ve already added information of the possible negative effect in the Conclusion part (Line 689-690).
Thank you again for your valuable comment.
- There is no real-world testing experiment with the distinguished kombucha sample KACL30-D7.
Response: Thank you very much for your valuable comment. Our study aimed to reveal the possibility for valorization of Cannabis leaves ̶ the waste from Cannabis plantation farm, which is legally permitted to use as food in Thailand with the limit of THC and CBD content per 1 serving size lower than 1.41 mg and 1.60 mg, respectively. For our final product, THC and CBD content were not analyzed due to the original content in the raw material (Cannabis leaves) was very low and do not exceed the level of Thai regulations. The best prototype product (KACL30-D7: Kombucha with 30% Cannabis leaves infusion and fermented for 7 days) was received as a model that can apply to another beverages. This sample was distinguished from another sample and was selected based on quality characteristics and the consumer liking score. In fact, the real-world testing on the KACL30-D7 kombucha sample through a sensory evaluation using the 9-point hedonic scaling test was performed in a controlled laboratory setting under the supervision of a sensory expert and thesis committee. To ensure relevant participant selection, only consumers with prior experience consuming cannabis leaves were included, as cannabis leaves is legally permitted in Thailand. However, detailed sensory data from our study will be published separately in another manuscript and cannot be included in this manuscript. If there is still further information that should include or discuss in our manuscript, please do not hesitate to let us know. Thank you very much again for your valuable comment.

Round 2
Reviewer 1 Report
Comments and Suggestions for Authors
All questions were clarified. I approve the publication of the manuscript.